# Value of Ultrasound Super-Resolution Imaging for the Assessment of Renal Microcirculation in Patients with Acute Kidney Injury: A Preliminary Study

**DOI:** 10.3390/diagnostics14111192

**Published:** 2024-06-05

**Authors:** Xin Huang, Yao Zhang, Qing Zhou, Qing Deng

**Affiliations:** Department of Ultrasound, Renmin Hospital of Wuhan University, Wuhan 430060, China; hxrmyy4896@whu.edu.cn (X.H.); 17612778397@163.com (Y.Z.)

**Keywords:** ultrasound, super-resolution imaging, acute kidney injury, microcirculation

## Abstract

The present study aimed to explore the clinical applicability of ultrasound super-resolution imaging (US SRI) for assessing renal microcirculation in patients with acute kidney injury (AKI). A total of 62 patients with sepsis were enrolled in the present study—38 with AKI and 24 control patients—from whom renal ultrasounds and clinical data were obtained. SonoVue contrast (1.5 mL) was administered through the elbow vein and contrast-enhanced ultrasound (CEUS) images were obtained on a Mindray Resona A20 ultrasound unit for 2 min. The renal perfusion time-intensity curve (TIC) was analyzed and, after 15 min, additional images were obtained to create a microscopic blood flow map. Microvascular density (MVD) was calculated and its correlation with serum creatinine (Scr) levels was analyzed. There were significant differences in heart rate, Scr, blood urea nitrogen, urine volume at 24 h, and glomerular filtration rate between the two groups (*p* < 0.01), whereas other characteristics, such as renal morphology, did not differ significantly between the AKI group and control group (*p* > 0.05). The time to peak and mean transit times of the renal cortex in the AKI group were prolonged compared to those in the control group (*p* < 0.01), while the peak intensity and area under the TIC were lower than those in the control group (*p* < 0.05). The MVD of the renal cortex in the AKI group was lower than that in the control group (18.46 ± 5.90% vs. 44.93 ± 11.65%; *p* < 0.01) and the MVD in the AKI group showed a negative correlation with Scr (R = −0.84; *p* < 0.01). Based on the aforementioned results, US SRI can effectively assess renal microcirculation in patients with AKI and is a noninvasive technique for the diagnosis of AKI and quantitative evaluation of renal microcirculation.

## 1. Introduction

Acute kidney injury (AKI) is a common syndrome, with an incidence rate as high as 20% among hospital inpatients, characterized by a rapid decline in renal function and the accumulation of metabolic waste products in the body [1]. According to statistics, the number of deaths caused by AKI exceeds 2 million per year worldwide; therefore, AKI has become a serious public health issue, greatly increasing medical and financial costs for patients’ families and society as a whole [2]. AKI has a variety of causes and complex mechanisms and is characterized by a high incidence rate, high mortality, and the ability to produce great harm [3]. Research has found that microcirculation disorders in renal tissue are an important cause of the occurrence and progression of AKI [4]. Therefore, the accurate assessment of renal microcirculation and timely intervention are key strategies for the prevention and treatment of AKI.

At present, multiple imaging techniques, such as enhanced computed tomography (CT) and magnetic resonance imaging (MRI), can be used to evaluate changes in the renal microvasculature and perfusion, although iodine-based contrast agents can exacerbate renal injury in patients with AKI. Ultrasound (US) imaging, however, has advantages over CT and MRI, such as safety, non-invasiveness, economic feasibility, and convenience. Whereas, traditional renal US was limited to evaluating the blood flow parameters of the branches above the renal interlobular artery, making it difficult to evaluate renal cortical blood flow. Emerging contrast-enhanced ultrasound (CEUS) techniques can be used to semi-quantitatively evaluate the renal microcirculation [5]. However, two-dimensional (2D)-US imaging approaches for the noninvasive evaluation of changes in the kidney microvasculature remain limited and cannot display tiny blood vessels, largely due to sonic diffraction.

To overcome this limitation, therefore, US super-resolution imaging (SRI) technology has emerged and has been utilized in both animal experiments and clinical research. SRI has extremely high temporal and spatial resolutions, enabling the identification of even the micron-level blood vessels and allowing for the noninvasive evaluation of renal microcirculation [6,7]. As such, the present study aimed to explore the feasibility and clinical applicability of US SRI for evaluating the microvasculature of patients with AKI, providing a new method with which to quantitatively evaluate renal microcirculation.

## 2. Materials and Methods

### 2.1. Participants

The present study included 62 patients with sepsis who were treated at the Renmin Hospital of Wuhan University between November 2023 and February 2024 [8]. Among these patients, 38 had sepsis-related AKI (AKI group) and 24 had sepsis but did not develop AKI (control group). The diagnosis of AKI was based on the Kidney Disease Improving Global Outcomes (KDIGO) criteria, as follows: (ⅰ) serum creatinine (Scr) increased by ≥26.5 μmol/L within 48 h; (ⅱ) Scr increased >1.5 times baseline within 7 days; or (ⅲ) urine output decreased (<0.5 mL/kg/h) for ≥6 h [9]. The exclusion criteria for the AKI group were as follows: (i) AKI stage 3–5; (ii) previous chronic kidney disease; and/or (iii) severe sepsis with shock or disseminated intravascular coagulation (DIC).

The inclusion criteria for the control group were (ⅰ) Scr within the normal range (male: 44–133 μmol/L, female: 70–106 μmol/L); (ⅱ) glomerular filtration rate (eGFR) was within the normal range (80–125 mL/min); and (ⅲ) no previous urinary disease and normal urological ultrasound examination. The protocol for the present study was approved by the Clinical Research Ethics Committee of Renmin Hospital of Wuhan University (No. WDRY2021-K115) and informed consent was obtained from all patients.

### 2.2. US Image Acquisition

All patients underwent 2D-US, color doppler flow imaging (CDFI), and contrast-enhanced US (CEUS) imaging using a Resona A20 Pro US unit (Mindray Bio-Medical Electronics Co., Ltd., Shenzhen, China) and an SC7-1U convex array transducer. The dynamic range, gain, and depth were tailored for the US machine and, to avoid motion during the scan, all patients were advised to hold their breath as instructed. Each kidney was examined by B-mode and CDFI was performed to evaluate blood flow throughout the kidney. During the examination, the patients were in a prone position, the same as for the follow-up examinations. 

The measurement parameters included the size, parenchymal thickness, cortical thickness, and interlobar artery flow resistance index (RI) of the kidney. Renal blood perfusion was evaluated using a semiquantitative grading scale, as follows: grade 0, no renal vessels detected; grade 1, a few vessels visible at the renal hilum; grade 2, interlobar arteries visible at the renal hilum and most of the renal parenchyma; and grade 3, renal vessels visible to the level of the arcuate artery.

### 2.3. CEUS Image Acquisition

For the standard CEUS examination, real-time dual-mode images (B-mode and CEUS) were used to guide the imaging plane and monitor microbubble signals following the injection. The imaging plane was kept constant to ensure that all images captured the same anatomy. The typical clinical CEUS examination was performed with 1.5 mL SonoVue (Bracco, Milan, Italy) microbubble solution injected as a bolus into the elbow vein, followed by 5 mL of normal saline. The mechanical index (MI) of 0.08 was utilized to prevent microbubble damage during CEUS exams. All of the pictures were saved in DICOM and AVI formats for further processing and the images were continuously collected for 2 min, with the renal cortex region in the middle pole of the kidney selected as the region of interest (ROI). Parameters relevant to the time-intensity curve (TIC) were obtained, including time to peak (TTP), mean transit time (MTT), peak intensity (PI), and area under the TIC (AUC). The analysis was repeated three times and the average values of the perfusion parameters were obtained.

### 2.4. US SRI Acquisition

By 15 min after the first CEUS, the contrast agent in the patient’s kidneys had almost completely disappeared. Subsequently, the patient was given a second injection of SonoVue (0.5 mL) through the elbow vein for a second scan and the MI was also 0.08. Using the microangiography mode of the US unit, the patients were instructed to hold their breath when the contrast agent intensity peaked, at which time the probe was kept stable to collect dynamic images for 4 s. The machine acquired 2000 image frames at a high frame rate of 500 frames/s and automatically generated the SRI of the renal blood vessels. Then, we chose three different renal cortex regions in the middle pole of the kidney to calculate the microvascular density (MVD), with internal software analysis of the US unit; thus, we obtained the average value.

### 2.5. Statistical Analysis

Continuous variables with a normal distribution were reported as mean ± standard deviation. Comparisons between the two groups were performed using an independent sample *t*-test, while non-normally distributed quantitative data were expressed as median and quartile (M [Q1, Q3]) using a non-parametric test. Count data were expressed as frequency or percentage (*n*, %) using the χ^2^ test. Pearson’s correlation analysis was used for correlation analysis and GraphPad Prism (version 8.0) was used for charting. Statistical significance was set at *p* < 0.05.

## 3. Results

### 3.1. Baseline Characteristics

Table 1 shows the baseline characteristics of all of the patients included in the present study. There were significant differences in the heart rate, Scr, blood urea nitrogen (BUN), urine volume at 24 h, and eGFR between the two groups (*p* < 0.01), while other characteristics did not differ significantly.

### 3.2. US and CEUS Parameters

Compared with the control group, there were no significant differences in the size, parenchymal thickness, cortical thickness, or blood perfusion of the kidneys in the AKI group; however, the RI of the interlobar arteries did increase (*p* < 0.01). The TIC parameters showed that the TTP and MTT in the AKI group were higher than those in the control group (*p* < 0.01), whereas the PI and AUC were lower than those in the control group (*p* < 0.01), as shown in Table 2.

### 3.3. US SRI Parameters

US SRI showed that the intrarenal blood vessels were distributed in a dendritic pattern directly beneath the renal capsule. Furthermore, the SRI displayed significantly improved spatial resolution (resolution of 100 μm) for renal cortical microvascular diameter (Figure 1). The microvascular density of the renal cortex in the AKI group (18.46 ± 5.90%) was significantly lower than that in the control group (44.93 ± 11.65%; *p* < 0.01); meanwhile, the MVD of the renal cortex in the AKI group was significantly negatively correlated with Scr (R = −0.84; *p* < 0.01), as shown in Figure 2.

## 4. Discussion

AKI is a common complication among severely sick patients and is often characterized by poor prognosis and high mortality [10]. Renal microcirculatory dysfunction and tissue hypoxia are pivotal pathophysiological mechanisms in the development and progression of AKI; however, there is, as of yet, no imaging technique that can noninvasively and sensitively detect renal microcirculation. Therefore, we urgently need new noninvasive detection methods.

At present, CEUS is another active imaging modality for the evaluation of AKI [5], as it is capable of detecting minute blood flow signals and displaying renal microcirculatory perfusion in real time, owing to the micron-sized microbubbles that can pass through the human microvasculature. As the results of the present study showed, interlobar artery RI increased in the AKI group, while other indicators, such as kidney size, parenchymal thickness, cortical thickness, and renal blood flow abundance (Table 2, Figure 1A,B), did not differ significantly. As such, we performed CEUS on all patients and analyzed the TIC parameters of the renal cortex and found that TTP and MTT were significantly increased, while the PI and AUC of the renal cortex in the AKI group were significantly decreased (Table 2, Figure 1C). These results indicated that there was no significant morphological abnormality or reduction in renal blood flow in patients with AKI; however, microcirculatory perfusion of the renal cortex was significantly decreased. Clinical studies and animal experiments have confirmed that AKI may occur even when renal blood flow is normal or even increased [11,12]. Therefore, intrarenal microcirculatory perfusion disorders are more sensitive and important than renal blood flow disorders in the occurrence and progression of AKI.

The severity of AKI in the renal cortex can be predicted using RT and MTT assays. It has been found, however, that the slopes of the cortical rise and PI correlate with AKI recovery [5]. Although CEUS can provide a semi-quantitative assessment of renal microcirculation, it cannot visually display the microvascular distribution of the renal parenchyma and requires software analysis. In recent years, however, owing to the great improvement in US equipment and inspired by optical SRI, US SRI has been investigated for its diagnostic and prognostic value for a variety of diseases. It utilizes the localization of ultrasound contrast agents within the blood vessels for the noninvasive display of the microvasculature. Additionally, related research has demonstrated the diagnostic value of US SRI in distinguishing between benign and malignant thyroid nodules or breast masses [13,14]. As for kidney diseases, US SRI can also provide a noninvasive assessment of renal microvasculature changes during AKI [15]. The core of the contrast agent microbubble is an inert gas that can be exhaled through the lungs without being excreted through the kidneys. Therefore, it is a more suitable choice than contrast media in CT or MRI examinations of severely ill patients without nephrotoxicity. Additionally, SRI can be performed at the bedside, making it more suitable for critically ill patients than CT or MRI [16].

US SRI, also called ultrasound localization microscopy (ULM) [17], can capture images and track the movement trajectory of microbubbles at an extremely high speed (500 frames/s) over a very short period (several seconds) [18], automatically generating microvascular images of living tissues. SRI technology can improve the resolution of blood vessels, with a spatial resolution of 100 μm, and the results of the present study showed that the tiny blood vessels near the renal capsule were clearly visible (Figure 1D,E). SRI can also provide a technical foundation for the intuitive and quantitative assessment of microcirculation in the renal parenchyma [19]. The MVD in the cortex was significantly reduced, indicating microcirculatory dysfunction in the renal cortex of patients with AKI (Figure 2A). Previous studies have shown damage to the endothelial cells of glomerular capillaries, extensive microvascular embolism, and neutrophil infiltration in the renal interstitium in patients with AKI [11], which may be the cause of the reduced MVD in the renal cortex.

Currently, a diagnosis of AKI relies primarily on Scr and urine volume; however, Scr is not a sensitive indicator of renal function. The Pearson correlation analysis results showed that the MVD in the renal cortex region was negatively correlated with Scr (Figure 2B), indicating that SRI technology can visually and quantitatively assess the severity of microcirculation disorders in the renal cortex of patients with AKI. This is consistent with previous studies showing that the occurrence of renal microcirculation disorders in the development of AKI occurred earlier than the increase in Scr [20]. Therefore, visually displaying renal cortex microcirculation using SRI technology may open new avenues for the early diagnosis of AKI.

There are some limitations to the present study. First, a larger sample size may be required to reach more definitive conclusions regarding US SRI in patients with AKI. Second, SRI generation is time-consuming and collecting the data for SRI datasets is difficult, as is true for all SRI investigations on humans. Finally, US examinations during follow-up visits should be performed to further understand the role of the US in predicting prognoses in patients with AKI. Further validation of our findings in a larger longitudinal study with a longer follow-up period is warranted.

## 5. Conclusions

In conclusion, when compared with US, CDFI, and CEUS, US SRI can display the features of renal microcirculation in AKI patients, with about 100-micron resolution. Meanwhile, US SRI paired with CEUS could be used as a good noninvasive imaging tool to assist in discriminating between AKI and no AKI in patients with sepsis. Based on these findings, we propose using US SRI as a noninvasive technique for the diagnosis of AKI and will provide more information for the treatment of patients with AKI as early as possible in the future.

## Figures and Tables

**Figure 1 diagnostics-14-01192-f001:**
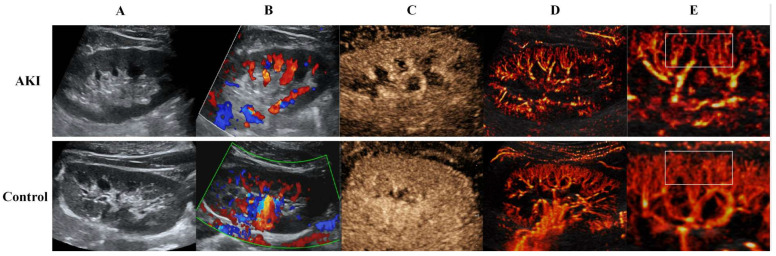
Images of patients with AKI and without AKI. (**A**,**B**): Two-dimensional-US image and CDFI of the kidneys. (**C**): CEUS of the kidneys reaching PI. (**D**): US SRI of the whole kidneys. (**E**): US SRI of the renal cortex with white squares.

**Figure 2 diagnostics-14-01192-f002:**
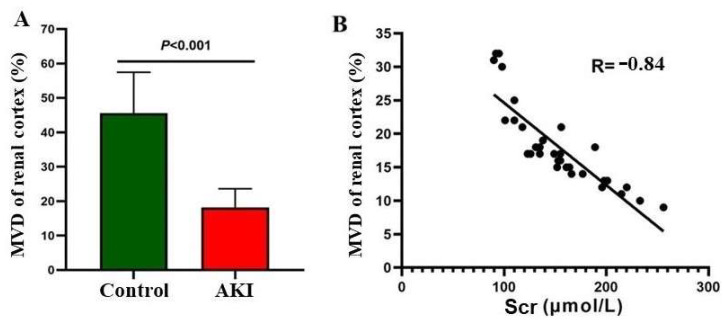
MVD of the renal cortex in two groups. (**A**): Comparison of the MVD of two groups. (**B**): Relationship diagram of MVD and Scr.

**Table 1 diagnostics-14-01192-t001:** Baseline characteristics of all patients.

	AKI Group	Control Group	*p*
	*n* = 38	*n* = 24	
Male gender, *n* (%)	17 (44.7%)	11 (45.8%)	0.892
Age, years, mean ± S.E.	62.4 ± 11.7	61.8 ± 12.1	0.864
**Complications, *n* (%)**			
Hypertension	23 (60.5%)	14 (58.3%)	0.748
Coronary heart disease	9 (23.7%)	6 (25.0%)	0.815
Diabetes	14 (36.8%)	8 (33.3%)	0.795
Cerebral apoplexy	6 (15.8%)	3 (12.5%)	0.563
Use of vasoactive drugs	14 (36.8%)	4 (16.7%)	0.073
**Heart function**			
Heart rate (BPM)	113.2 ± 16.5	86.2 ± 13.4	<0.001
Systolic pressure (mmHg)	126.7 ± 21.0	121.5 ± 19.8	0.593
Diastolic pressure (mmHg)	77.4 ± 15.3	79.3 ± 14.2	0.772
Left ventricular systolic function (%)	64.2 ± 6.1	66.3 ± 5.7	0.648
**Renal function**			
Scr (μmol/L)	152.7 ± 44.1	81.5 ± 19.1	<0.001
BUN (μmol/L)	13.4 (9.5, 18.4)	5.3 (4.8, 6.9)	<0.001
Urine volume of 24 h (mL)	347 (271, 649)	1463 (1283, 1895)	<0.001
eGFR (mL/min)	59.3 ± 14.7	87.2 ± 4.8	<0.001

**Table 2 diagnostics-14-01192-t002:** Two-dimensional-US and CEUS parameters of all patients.

	AKI Group	Control Group	*p*
	*n* = 38	*n* = 24	
**2D-US parameters**			
Renal size (mm)	114.6 ± 12.4	113.7 ± 13.6	0.712
Renal parenchyma thickness (mm)	14.6 ± 1.9	14.3 ± 2.1	0.518
Renal cortical thickness (mm)	8.2 ± 1.2	7.9 ± 1.1	0.273
**Renal blood perfusion**			
Grade 2, *n* (%)	4 (10.5)	2 (8.3)	0.832
Grade 3, *n* (%)	34 (89.5)	22 (91.7)	
RI of the interlobar artery	0.69 ± 0.09	0.61 ± 0.08	0.003
**CEUS parameters**			
TTP (s)	47.1 ± 9.9	35.54 ± 5.48	<0.001
MTT (s)	80.3 ± 12.4	54.2 ± 9.1	<0.001
PI (dB)	33.9 ± 5.5	43.1 ± 4.6	<0.001
AUC (dB)	3198.9 ± 778.5	4348.5 ± 539.8	<0.001

## Data Availability

The datasets used and analyzed in the current study are available from the corresponding author upon reasonable request.

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
