# Peer review of "Value of Ultrasound Super-Resolution Imaging for the Assessment of Renal Microcirculation in Patients with Acute Kidney Injury: A Preliminary Study"

_diagnostics, 2024, doi:10.3390/diagnostics14111192_

Round 1
Reviewer 1 Report
Comments and Suggestions for Authors
The authors present a nice short study to assess the power of US SRI in AKI diagnosis. They present sample images from US, doppler, CEUS and US SRI images and compare it to levels of Scr. They find that microvascular density (MVD) highly correlates with Scr levels, an exciting result. I think this is well suited for Diagnostics with only some minor suggestions.
Line 42: The most advanced examples of US SRI have just been benchmarked in a paper by Lerendegui et al. https://ieeexplore.ieee.org/abstract/document/10497610 Please consider citing this recent (2024) survey of the technique.
Section 2.4: I am interested in this automatic US SRI capture. Can the authors (briefly) elaborate on what type of processing is happening here? What are the frequency and MI used? These are certainly beautiful US SRI images (Fig. 1).
I am curious if the Doppler signal would also have been correlated to Scr. Do the authors have this data?
Line 155: I’m not sure whether the authors desire to add a sentence or two on how microbubbles work (if you assume the readers aren’t familiar with US CEUS), but if the authors want to here would be a good place to do so. Basic references to review articles (as an example: Yusefi et al. Frontiers in Physics 2022) would be fine.
Reviewer 2 Report
Comments and Suggestions for Authors
In the present study is analysed the clinical applicability of ultrasound super-resolution imaging (US SRI) for assessing renal microcirculation in patients with acute kidney injury.
The topic of the paper is very interesting, the structure of the paper is well organized and well written, and it is easy to follow.
In the reviewer opinion, the present paper meets all requirements for the published in Diagnostics journal after a minor revision.
Comments:
1. Page 2, line 91
MVD is a parameter given directly from the machine after the complete set of acquisitions? Please clarify.
2. Page 3, line 101
Table 1 must be introduced in the text before fig 1 for better reading ease.
3. Page 3, line 109
Same comment as previous for table 1.
4. Page 3, line 114
Comment if there were any criteria related to the patient to present the selected images in the Fig. 1.
5. Page 6, line 186
Submicron means less the one micron. Please correct.
